# An Optimal Vehicle-Scheduling Model for Signal-Free Intersections Considering Bus Priority in a Connected and Automated Vehicle Environment

**DOI:** 10.3390/s25175438

**Published:** 2025-09-02

**Authors:** Dongliang Wang, Shunjie Jiang, Guorong Zheng, Xiaohu Shi

**Affiliations:** 1College of Software, Jilin University, Changchun 130012, China; wangdl1621@mails.jlu.edu.cn (D.W.); jiangsj5522@mails.jlu.edu.cn (S.J.); 2School of Electrical and Control Engineering, North China University of Technology, Beijing 100144, China; 3College of Computer Science and Technology, Jilin University, Changchun 130012, China

**Keywords:** automated vehicles, conflict area analysis, speed guidance, optimal scheduling, bus priority

## Abstract

The optimal scheduling of vehicles at signal-free intersections under the connected and automated vehicle (CAV) environment has become a research hotspot in the intelligent transportation field. However, existing studies often oversimplify the intersection’s conflict area and fail to adequately address the spatiotemporal sparsity of conflict points, with little attention given to bus priority requirements. To address these gaps, this paper first establishes an intersection coordinate system and constructs a conflict area analysis model based on the coordinates of key conflict points and vehicle trajectories. Subsequently, an optimal scheduling model for automated vehicles at signal-free intersections with bus priority is developed, which considers the set of vehicles influencing decisions within a time window and uses vehicle entry times and lateral lane changes as decision variables. To enhance computational speed while preserving convergence accuracy, a search space reduction method based on available gaps for conflict point traversal constraints is designed. The model is then solved using an improved double-layer multi-population particle swarm optimization (PSO) algorithm. Simulation results, compared against traditional signal control, rule-driven signal-free, and dynamic-optimization-based signal-free algorithms demonstrate that the proposed method achieves a favorable balance between computational cost and efficiency. It significantly reduces the average vehicle delay. Moreover, incorporating bus priority reduces the average per capita delay by 18.95% compared to the non-priority scenario, effectively proving the validity of the proposed method.

## 1. Introduction

At conventional signalized intersections, vehicle passage is primarily dictated by traffic light indications or by drivers’ decisions based on their personal experience and judgment of the surrounding environment. Although signalized intersections can effectively reduce the number of conflicts, their fixed signal cycles often lead to unnecessary waiting times as vehicles decelerate or stop, sometimes even causing congestion and thereby reducing the intersection’s overall traffic efficiency [1].

With the rapid development of connected vehicle and autonomous driving technologies, signal-free intersection control has garnered increasing attention from researchers. How to guide connected and automated vehicles (CAVs) to pass through signal-free intersections safely and efficiently has become a prominent research topic in the field of intelligent transportation systems.

On this issue, Naumann et al. [2] proposed a method that identifies potential conflicts by predicting vehicle trajectories and avoids collisions by adjusting their right-of-way, providing a reference for developing passage strategies for CAVs in mixed-traffic environments. Dresner et al. [3,4] introduced the early Autonomous Intersection Control (AIC) model, which spurred subsequent research by scholars worldwide to further enhance its efficiency.

Fayazi et al. [5] proposed a vehicle-scheduling method based on Mixed-Integer Linear Programming (MILP). This method uses an intersection control server to process data streams from automated vehicles and assigns optimal arrival times to CAVs to minimize stops and intersection delays, with the optimization solved by IBM CPLEX. Simulation results showed that this scheme significantly improved intersection efficiency compared to baseline methods and could be extended to mixed-traffic environments. Ahn et al. [6] proposed a CAV control algorithm for signal-free intersections based on the job-shop scheduling problem, which optimizes the timing for vehicles entering conflict zones while ensuring safety, with its effectiveness and real-time performance validated by simulations. Li et al. [7] employed wireless communication and a genetic algorithm to search for the optimal vehicle passing sequence and trajectories, simultaneously optimizing system performance and vehicle paths. Kong et al. [8] formulated a MINLP model for CAV trajectory optimization, which maximized average speed by linearizing trajectories to control the timing of CAVs passing through conflict points, with constraints introduced after analyzing these points. Numerical experiments verified the method’s ability to enhance intersection safety and efficiency. Chai et al. [9] presented an intersection coordination method for CAVs based on pre-assigned time slots. The method calculates target states in a state-adjustment zone and optimizes the driving sequence and operational advice using LOOSE and COMPACT methods, enabling CAVs to traverse the intersection safely at optimal speeds and improving capacity. Chen et al. [10] developed a reservation-based model with a controllable gap strategy, designing speed-dependent safety constraints to optimize vehicle gaps and reservation sequences. Simulation results indicated that the model achieved local rapid decision-making and global conflict resolution, thereby enhancing intersection efficiency. To reduce computational complexity and ensure real-time solvability, some researchers have focused on improving efficiency through platooning or queuing optimization. Medina et al. [11] considered vehicle dynamics in the AIC model and introduced the concept of virtual platoons to control vehicles from different lanes and directions as queues. Zhou et al. [12] proposed a cooperative intersection control method based on virtual platooning, using a Distributed Adaptive Sliding Mode Controller (DASMC) to coordinate platoon passage and combining it with a traffic flow regulation mechanism to optimize traffic order, effectively alleviating congestion and intersection spillover. Gao et al. [13] introduced a Dynamic Resequencing and Platooning (DRP) strategy, which optimized the passing sequence of CAVs based on a state-transition network and used a decentralized energy-optimal control framework to optimize trajectories, significantly reducing intersection delay and energy consumption. Hu et al. [14] proposed a multi-vehicle cooperative method based on virtual platoons and a constraint tree. This approach first maps approaching vehicles onto virtual lanes and then constructs a constraint tree to determine a conflict-free passing sequence, forming local virtual platoons, with hierarchical robust control ensuring vehicles pass safely in order. Chen et al. [15] presented a graph theory-based conflict resolution method. By constructing a directed conflict graph and an undirected coexistence graph to describe vehicle conflict relationships, they solved for local and global optimal passing sequences using a depth-first spanning tree and a maximum matching algorithm, respectively, thereby effectively improving overall intersection efficiency.

Some researchers have adopted distributed control methods to schedule CAVs at signal-free intersections to alleviate the burden on computational resources. Zhang et al. [16] proposed a distributed optimal control method that enables autonomous and optimized passage of CAVs through signal-free intersections while ensuring collision avoidance and safe following distances within the control zone. Bian et al. [17] introduced a distributed optimization and control method that decomposes the cooperative passage problem into three parts: state observation, arrival time optimization, and trajectory tracking control; using a distributed algorithm for fixed-time observation and scheduling to achieve vehicle cooperation without global coordination. Yu et al. [18] proposed a distributed hierarchical optimization and control architecture that uses Distributed Model Predictive Control (DMPC) to optimize traffic efficiency at the intersection level and energy savings for individual platoons, with simulations verifying its effectiveness in both aspects. Xu et al. [19] constructed a MINLP-based trajectory optimization model for CAVs. By analyzing potential conflict points between vehicles, the model establishes kinematic and collision avoidance constraints to guide vehicles safely and efficiently, with simulation results validating its effectiveness in improving average speed and throughput. Cheng et al. [20] proposed a game-theory-based conflict resolution model that combines cooperative and non-cooperative decisions to optimize multi-vehicle conflict relationships, enhancing intersection stability and robustness and reducing average vehicle delay by about 5%. Xu et al. [21] developed a V2X-based management framework for signal-free intersections, using a resource reservation model to avoid collisions and reinforcement learning to optimize scheduling, proposing a function approximation-based control algorithm (FA-NIC) to improve scheduling efficiency and traffic strategies. Belkhouche [22] proposed a Lagrange function-based cooperative conflict resolution strategy for vehicles, solving for optimal avoidance schemes from both cost and time perspectives, with its effectiveness validated by simulations. Liu et al. [23] optimized straight-going vehicle decisions using reinforcement learning and speed prediction, balancing safety and efficiency through collision critical point calculations and desired speed guidance. Peng et al. [24] proposed a cooperative decision-making framework for CAVs that considers personalized driving behaviors, combining motion prediction and game theory to optimize conflict resolution. Hardware-in-the-loop tests verified that the algorithm enhances driving safety and exhibits good real-time and computational performance. Li et al. [25] proposed the COOR-PLT hierarchical control model based on deep reinforcement learning, which dynamically forms platoons via a centralized control strategy and optimizes the passing sequence using a decentralized strategy, thereby improving intersection throughput and reducing fuel consumption.

An analysis of the aforementioned research reveals the following gaps:Existing studies mostly conduct vehicle conflict analysis for standard intersections. However, variations in intersection geometry can alter the spatial critical points of conflict between vehicles.This is a typical NP-hard problem characterized by the spatiotemporal sparsity of conflict points. How to strike a balance between convergence accuracy and computational speed to achieve an optimal trade-off between the optimization results and algorithmic efficiency requires further exploration.Existing methods have seldom focused on bus priority demands. In an automated driving environment, the need for bus priority persists and gives rise to a synergistic optimization problem of spatial and temporal road resources, which in turn affects existing optimal scheduling strategies for automated vehicles.

Based on this, this paper investigates an optimal vehicle-scheduling model for signal-free intersections that considers bus priority in a connected and automated vehicle environment. The main contributions of this paper are as follows:A conflict area analysis model based on the coordinates of key conflict points and vehicle trajectories is constructed, grounded in an intersection coordinate system.The model specifically addresses the demand for bus priority on dedicated lanes at signal-free intersections in a CAV environment, establishing an optimal scheduling model for automated vehicles that uses vehicle entry times and lateral lane changes as decision variables.To enhance computational speed while maintaining convergence accuracy, a search space reduction method based on available gaps for conflict point traversal constraints is designed. Furthermore, an improved double-layer multi-population particle swarm optimization algorithm is proposed to solve the model.

This paper is organized as follows: Section 2 presents the conflict area analysis, constructing the model based on key conflict points and vehicle trajectories. Section 3 details the construction of the optimal vehicle control model for signal-free intersections considering bus priority in an automated driving environment. Section 4 introduces the search space reduction method. Section 5 provides the model’s solution, validation, and a comparative analysis.

The notations used for the following sections are listed in Table 1.

**Table 1 sensors-25-05438-t001:** Notations.

Notation	Definition
$x_{dir,j},y_{dir,j}$	The x and y coordinates of the endpoints for each inbound lane at the intersection.
dir	Inbound direction of the intersection.
$n_{dir}$	Number of inbound lanes for direction *dir*.
$X_{fdir,j},Y_{fdir,j}$	The x and y coordinates of the endpoints for each outbound lane at the intersection.
i,j	Lane ID.
n	Total number of inbound approaches.
fdir	Outbound direction of the intersection.
$n_{fdir}$	Number of outbound lanes for direction *fdir.*
m	Total number of outbound approaches.
$x_{a},y_{a}$	The x and y coordinates of point *a*.
ΔT	Control time window.
$T_{safe}$	Safe time headway.
$a_{\max}$	Maximum acceleration of the vehicle.
$b_{\max}$	Maximum deceleration of the vehicle.
$v_{\max}$	Maximum speed limit of the road section.
$v_{\min}$	Minimum guidance speed of the road section.
$L_{c}$	Length of the road section’s control zone.
$p_{dir,i}^{k}(t_{0})$	Position of the *k*-th vehicle on inbound lane *i* in direction *dir* at the start of guidance.
$v_{dir,i}^{k}(t)$	Speed of the *k*-th vehicle on inbound lane *i* in direction *dir* at time *t*.
$t_{dir,i}^{k}$	The time at which the *k*-th vehicle on inbound lane *i* in direction *dir* enters the intersection.
$l_{dir,i}^{k}$	Distance of the *k*-th vehicle on inbound lane *i* in direction *dir* from the stop line at the start of guidance.
$a_{c}$	Comfortable acceleration of the vehicle.
$a_{dir,i}^{k}$	Uniform acceleration under speed guidance.
$S_{dir,i}^{k}$	Distance traveled by the *k*-th vehicle on inbound lane *i* in direction *dir* when it reaches the guidance speed.
$t_{dir,i}^{k,b}$	The time at which the last automated vehicle ahead of bus *k* on inbound lane *i* in direction *dir* enters the dedicated lane.
$t_{dir,i}^{k,e}$	The time at which bus *k* on inbound lane *i* in direction *dir* exits the intersection.
$a_{dir,i}^{k,bus}(t)$	Acceleration of bus *k* on inbound lane *i* in direction *dir* at time *t*.
$v_{dir,i}^{k,bus}(t)$	Speed of bus *k* on inbound lane *i* in direction *dir* at time *t*.
$d_{dir_{1},dir_{2},i,j}^{k}$	Delay time of the *k*-th vehicle traveling from lane *i* of inbound approach *dir*1 to lane *j* of outbound approach *dir*2 within the time window.
$t_{0}$	Start time of guidance.
$\mu_{1}$	Per capita delay conversion coefficient for general vehicles.
$\mu_{2}$	Per capita delay conversion coefficient for buses.
$C_{k}$	Vehicle type; 0 for a passenger car, 1 for a bus.
$K_{dir_{1},dir_{2},i,j}$	Number of vehicles traveling from lane *i* of inbound approach *dir*1 to lane *j* of outbound approach *dir*2 within the time window.
$t_{b,d^{+},c}^{k}$	Arrival time of vehicle *k* from direction *d*^+^ at conflict point *c*.
$t_{e,d^{+},c}^{k}$	Departure time of vehicle *k* from direction *d*^+^ from conflict point *c*.
$l_{d^{+},c,p}$	Distance from conflict point *c* to conflict point *p* in direction *d*^+^.
$l_{b,d^{+},1}^{k}$	Distance of vehicle *k* + 1 to the first conflict point at the beginning of the time window.
$l_{d^{+},1}^{k+1}(t)$	Distance of vehicle *k* + 1 to the first conflict point at time *t*.
$a_{d^{+},1}^{k}$	Guidance acceleration for vehicle *k* corresponding to conflict point 1 in direction *d*^+^.
$l_{h^{+},c}$	Distance from the stop line to conflict point *c* in direction *h*^+^.

## 2. Intersection Conflict Area Analysis

### Assumptions

To establish the foundation for the model, the following assumptions are made:Automated vehicles are not permitted to stop within the intersection area and must maintain a constant speed throughout their traversal.The intended exit direction for each automated vehicle is known in advance.The system operates under fully automated driving conditions, and communication delays for Vehicle-to-Vehicle (V2V) and Vehicle-to-Everything (V2X) are considered negligible.

As shown in Figure 1, the coordinates of the stop line endpoints for each approach lane are defined according to a clockwise convention:

Following the clockwise direction, the coordinates of the endpoints for each inbound lane are$$\left(x_{dir,j},y_{dir,j}),j=0,1,2,\ldots ,n_{dir},dir\in \{dir_{1},dir_{2},\ldots ,dir_{n}\right\}$$
where dir represents the inbound direction, $n_{dir}$ is the number of inbound lanes for the direction dir, and n is the total number of inbound approaches. For a standard four-approach intersection, for example, dir∈{North,South,West,East}.

Similarly, following the clockwise direction, the coordinates for the endpoints of each outbound lane are$$\left(X_{fdir,j},Y_{fdir,j}),j=0,1,2,\ldots ,n_{fdir},fdir\in \{fdir_{1},fdir_{2},\ldots ,fdir_{m}\right\}$$
where fdir represents the outbound direction, $n_{fdir}$ is the number of outbound lanes for the direction fdir, and m is the total number of outbound approaches.

Taking the trajectory conflict area of two straight-moving vehicles as an example, it can be seen from Figure 2 that the true conflict area between the two vehicles is not the region enclosed by a−b−c−d. When the rear of vehicle M has passed the position represented by the line segment ab in Figure 2, and the front of vehicle N has not yet passed the position represented by the line segment ad, the two vehicles will not collide, even though they are both within the geometric conflict area of their trajectories. Similarly, if the rear of vehicle N has passed the position represented by the line segment bc, and the front of the former vehicle M has not yet passed the position represented by the line segment cd, the two vehicles will also not collide.

Based on the analysis above, the calculation process for the non-conflicting condition between two vehicles is as follows: Let the coordinates of the inbound stop line endpoint and the outbound lane endpoint corresponding to the lane boundary line passing through the two points a and d be $\left(x_{dir_{1},i_{1}},y_{dir_{1},i_{1}}\right)$ and $\left(X_{fdir_{1},i_{2}},Y_{fdir_{1},i_{2}}\right)$, respectively, and let the coordinates of the inbound stop line endpoint and the outbound lane endpoint corresponding to the lane boundary line passing through the two points a and b be $\left(x_{dir_{2},j_{1}},y_{dir_{2},j_{1}}\right)$ and $\left(X_{fdir_{2},j_{2}},Y_{fdir_{2},j_{2}}\right)$, respectively. Then, the coordinates of point a, $\left(x_{a},y_{a}\right)$, can be calculated by simultaneously solving Equations (1)–(3):(1)$$\begin{array}{l} y=k_{1}x+b_{1}\\ \left\{\begin{array}{l} k_{1}=\frac{Y_{fdir_{1},i_{2}}-y_{dir_{1},i_{1}}}{X_{fdir_{1},i_{2}}-x_{dir_{1},i_{1}}}\\ b_{1}=y_{dir_{1},i}-\frac{Y_{fdir_{1},i_{2}}-y_{dir_{1},i_{1}}}{X_{fdir_{1},i_{2}}-x_{dir_{1},i_{1}}}x_{dir_{1},i_{1}} \end{array}\right. \end{array}$$(2)$$\begin{array}{l} y=k_{2}x+b_{2}\\ \left\{\begin{array}{l} k_{2}=\frac{Y_{fdir_{2},j_{2}}-y_{dir_{2},j_{1}}}{X_{fdir_{2},j_{2}}-x_{dir_{2},j_{1}}}\\ b_{2}=y_{dir_{2},m}-\frac{Y_{fdir_{2},j_{2}}-y_{dir_{2},j_{1}}}{X_{fdir_{2},j_{2}}-x_{dir_{2},j_{1}}}x_{dir_{2},j_{1}} \end{array}\right. \end{array}$$(3)$$\left\{\begin{array}{l} y=k_{1}x+b_{1}\\ y=k_{2}x+b_{2} \end{array}\right.$$

Similarly, the coordinates of point c can be calculated as $\left(x_{c},y_{c}\right)$.

To ensure that two vehicles from different directions do not conflict within the intersection, the following conditions must be satisfied: at time t, the coordinate of corner 1 of vehicle N (where corners are numbered clockwise starting from the top-left of the vehicle) is less than the coordinate of point a, while simultaneously the coordinate of corner 4 of vehicle M is greater than the coordinate of point a; or, at time t, the coordinate of corner 2 of vehicle M is less than the coordinate of point c, while simultaneously the coordinate of corner 3 of vehicle N is greater than the coordinate of point c. Therefore, the constraints are as shown in Equation (4).(4)$$\left\{\begin{array}{l} x_{N_{1}}(t)<x_{a}\\ y_{N_{1}}(t)<y_{a}\\ x_{M_{4}}(t)>x_{a}\\ y_{M_{4}}(t)>y_{a} \end{array}\right.or\left\{\begin{array}{l} x_{N_{3}}(t)>x_{c}\\ y_{N_{3}}(t)>y_{c}\\ x_{M_{2}}(t)<x_{c}\\ y_{M_{2}}(t)<y_{c} \end{array}\right.$$

It can be proven that if the aforementioned condition is met, vehicles M,N will not collide in the conflict area a−b−c−d during their subsequent travel. This is because if a collision were to occur, the distance traveled by vehicle M in a given time (taking the case where vehicle M enters the conflict zone later as an example) would need to be equal to the distance traveled by vehicle N divided by cosα, where α is the angle between the boundary lines bc and dc. Since both vehicles have the same speed, v, and 0<α<π, the aforementioned conclusion holds. As can be seen from Figure 2, if the angle between directions $dir_{1}$ and $dir_{2}$ is greater than π/2, it is evident that the two vehicles will also not collide when constraint (19) is satisfied.

A further analysis of the conflict area between a straight trajectory and a curved trajectory is performed, as shown in Figure 3. In this case, the trajectory of vehicle M is a curve, while the trajectory of vehicle N is a straight line. To ensure that no collision occurs between the vehicles, it is necessary that at time t, no point on the line segment representing the rear of vehicle M collides with corner 1 of vehicle N. Since the trajectory of vehicle M is a circular arc, to meet the above condition, the arc length traversed by each point on the rear line of vehicle M along its corresponding circular path in a given time must be greater than the arc length of the conflict zone between the straight lines ae and aq.

From Figure 3, it can be known that se=fq. Since the area of the sector oeq is less than the area of the triangle Δofq, it follows that r×arc eq/2<r×es/2, which implies that arc eq<es=fq.

Furthermore, in the conflict between a straight trajectory and a curved trajectory at an intersection, in general, $∠eaq<\frac{\pi }{4}$, and thus aq>fq, and since ae+oa>r=aq+oa, it follows that ae>aq. This further implies that  arc eq<ae. Therefore, in Figure 3, for vehicle M and vehicle N to avoid a collision, they only need to satisfy the following constraints as shown in Equation (5):(5)$$\left\{\begin{array}{l} x_{N_{1}}(t)<x_{a}\\ y_{N_{1}}(t)<y_{a}\\ x_{M_{4}}(t)>x_{a}\\ y_{M_{4}}(t)>y_{a} \end{array}\right.$$

Other cases can be deduced in a similar manner and will not be elaborated upon here due to space limitations.

## 3. Control Model Formulation and Method Design

In this paper, the inputs to the control model include the geometric parameters of the intersection, such as the number of inbound and outbound lanes for each direction, the coordinates of the lane stop line endpoints, the attributes of a vehicle’s approach lane (including whether it is a dedicated bus lane and its permitted movements, e.g., straight-through, left-turn, or a combination), and the attributes of the trajectory boundary (e.g., whether it is a bus trajectory). Additionally, the inputs include vehicle state information at the decision-making instant, such as position (both lateral and longitudinal) and speed. The outputs of the control model are the ID of any lane-changing vehicle and its target lane, the intersection entry and exit times for each vehicle, and the vehicle delay time.

The optimal scheduling is applied to the set of vehicles that enter the road section’s control zone (as shown in Figure 4) within a given time window ΔT. To ensure that all vehicles in this set can enter the intersection at a prescribed speed v, the size of the time window must satisfy the condition presented in Equation (6).(6)$$\left\{\begin{array}{l} \Delta T>T_{safe}\\ \Delta T\leq \min\left\{\frac{L_{c}-\frac{v_{\max}^{2}-v^{2}}{2b_{\max}}}{v_{\max}},\frac{L_{c}-\frac{v^{2}-v_{\min}^{2}}{2a_{\max}}}{v_{\min}}\right\} \end{array}\right.$$

The constraints for the optimization model are presented as follows:


**(1) Speed Guidance Constraint**


After a vehicle enters the control zone, it undergoes uniform acceleration or deceleration to reach the prescribed guidance speed for the intersection. Throughout this process, the time headway between the guided vehicle and its preceding vehicle must be maintained above the safe time headway. This condition is formally expressed as the constraint in Equation (7).(7)$$\left\{\begin{array}{l} p_{dir,i}^{k-1}(t_{0})+\int_{t_{0}}^{t}v_{dir,i}^{k-1}(\tau )d\tau \geq p_{dir,i}^{k}(t_{0})+\\ \int_{t_{0}}^{t}v_{dir,i}^{k}(\tau )d\tau +v_{dir,i}^{k}(t)T_{safe},\forall dir,k,i\\ t_{0}<t<t_{dir,i}^{k}\\ v_{\min}<v_{dir,i}^{k}(t)<v_{\max},\forall k,i \end{array}\right.$$

From this, it can be understood that the optimal scheduling of vehicles entering the control zone within the time window ΔT is, in effect, a control of the vehicles’ intersection entry time, $t_{dir,i}^{k}$. In a practical scenario, if the guidance speed on the road section is very low, it can be converted into the time elapsed from the start of guidance to exiting the intersection, thereby obtaining the intersection exit time. After this conversion, vehicles can travel freely and queue normally if they encounter a queue. However, their speed when departing from the stop line must reach the prescribed speed v, and the departure time must be consistent with the optimization result. Therefore, if a vehicle needs to stop and wait before entering the intersection and is the first vehicle in the queue on its lane, its stopping position should be at a certain distance from the stop line. This distance, denoted as $l_{dir,i}^{k,s}$, is the distance the vehicle can travel in the time required to accelerate from a standstill to the speed v. The time at which the vehicle starts from this stopping position is denoted as $t_{dir,i}^{k,s}$. The formulas for calculating $t_{dir,i}^{k,s}$ and $l_{dir,i}^{k,s}$ are shown in Equation (8).(8)$$\left\{\begin{array}{l} \frac{v-v_{dir,i}^{k}(t_{0})}{a_{dir,i}^{k}}+\frac{l_{dir,i}^{k}-S_{dir,i}^{k}}{v}=t_{dir,i}^{k,s}+\frac{v}{a_{c}}\\ S_{dir,i}^{k}=\frac{v^{2}-{\left(v_{dir,i}^{k}(t_{0})\right)}^{2}}{2a_{dir,i}^{k}}\\ l_{dir,i}^{k,s}=\frac{v^{2}}{2a_{c}} \end{array}\right.$$


**(2) Bus Priority Constraint**


In this paper, buses are also assumed to be automated vehicles that travel in dedicated bus lanes. To improve the utilization of spatial resources on these dedicated lanes, automated vehicles from adjacent general-purpose lanes can be controlled to enter the bus lane. If absolute bus priority is implemented at the signal-free intersection, it must be ensured that throughout the dedicated-lane bus’s passage through the intersection, its acceleration must satisfy the condition $a_{dir,i}^{k,bus}(t)\geq 0$ (i.e., the bus is never guided to decelerate), and that other automated vehicles entering the bus lane do not affect the bus’s speed. Furthermore, it is also required that when the bus’s acceleration is zero, its speed must be equal to the maximum speed (the presence of bus stops is not considered in this scenario). These conditions are formally expressed in Equation (9).(9)$$\left\{\begin{array}{l} a_{dir,i}^{k,bus}(t)\geq 0,t_{dir,i}^{k,b}\leq t\leq t_{dir,i}^{k,e}\\ v_{dir,i}^{k,bus}(t)=v_{\max},\forall a_{dir,i}^{k,bus}(t)=0,t_{dir,i}^{k,b}\leq t\leq t_{dir,i}^{k,e} \end{array}\right.$$

In this paper, the free-flow speed of buses and general vehicles is assumed to be the same, denoted by $v_{\max}$.


**(3) Lane-Changing Trajectory Constraint**


The lane-changing maneuvers considered in this paper include those performed by automated vehicles between general-purpose lanes, as well as lane changes into the dedicated bus lane. This study adopts the method presented in Reference [26] for planning the lane-changing trajectories of automated vehicles.


**(4) Set of Vehicles Affecting the Optimization**


Vehicles from the previous time window that have not yet passed through the intersection will impose constraints on the optimal control of vehicles in the current time window. Therefore, when performing the optimization for the current time window, these vehicles must be taken into account. The set of vehicles to be considered must satisfy the condition shown in Equation (10) (using a standard four-approach intersection as an example).(10)ifYkz(t)>Yfdir,j,z=3,4andfdir=NorthorYkz(t)<Yfdir,j,z=3,4andfdir=SouthorXkz(t)>Xfdir,j,z=3,4andfdir=Eastor Xkz(t)<Xfdir,j,z=3,4andfdir=Westthencdir,fdir,ijk∈CΔT
where $X_{k_{z}}(t)$ represents the x coordinate of corner z of vehicle k at time t, and $Y_{k_{z}}(t)$ represents the y coordinate of corner z of vehicle k at time t; $c_{dir,fdir,ij}^{k}$ represents the k−th vehicle traveling from lane i of the inbound approach in direction dir to lane j of the outbound approach in direction fdir; and $C_{\Delta T}$ represents the set of vehicles within the time window ΔT.


**(5) Conflict Constraints**


See the analysis in the Section 2 above for details.


**(6) Objective Function**


At the beginning of each time window, a two-stage optimization is performed for the lane changing and speed guidance of automated vehicles. The decision variables are $t_{dir,i}^{k}$ and $c_{dir,i,j}^{\Delta T}$, where $c_{dir,i,j}^{\Delta T}$ is the ID of the automated vehicle changing from lane i in direction dir to lane j within the time window ΔT. First, the lane-changing vehicles are determined, and then their guidance speeds are determined. These two stages are decided only once at the start of the time window. The decision process is influenced by the previous time window, but the set of vehicles being optimized does not include those that were already scheduled in the previous window.

When optimizing the signal-free intersection with bus priority, vehicle delay is converted into per capita delay. The delay for an individual vehicle is defined as the difference between its actual travel time (from entering the road section’s control zone until exiting the intersection) and its travel time at free-flow speed. The optimization aims to minimize the total per capita delay, which serves as the performance evaluation metric. The formulas for calculating the objective function are presented in Equations (11)–(13).(11)$$d_{dir_{1},dir_{2},i,j}^{k}=t_{dir_{1},i}^{k}-t_{0}-\frac{L_{c}}{v_{\max}}$$(12)$$D_{dir_{1},dir_{2},i,j}^{k}=\left\{\begin{array}{l} \mu_{1}d_{dir_{1},dir_{2},i,j}^{k},C_{k}=0\\ \mu_{2}d_{dir_{1},dir_{2},i,j}^{k},C_{k}=1 \end{array}\right.$$(13)$$\min D_{sum}=\sum_{dir_{1}=N,S,W,E}\sum_{dir_{2}=N,S,W,E}\sum_{i=1}^{I_{dir_{1}}}\sum_{j=1}^{J_{dir2}}\sum_{k=1}^{K_{dir_{1},dir_{2},i,j}}D_{dir_{1},dir_{2},i,j}^{k}$$

## 4. A Search Space Reduction Method for Conflict Point Traversal Constraints Based on Available Gaps

Directly solving the aforementioned optimization problem results in a computation time that grows exponentially with the number of vehicles [27,28]. Therefore, this paper proposes a search space reduction method to significantly reduce the computational load required during the optimization process.

First, the concept of the “minimum available gap” is defined. The time required for a vehicle, traveling at its guidance speed, to pass through a conflict point is given as shown in Equation (14).(14)$$\Delta T_{c}=\frac{l+w}{v}$$
where l is the length of the inserting vehicle and w is the width of the conflict point, which can be calculated according to Figure 2 and Figure 3. Therefore, the minimum available gap must satisfy Equation (15).(15)$$t_{b,j^{+},c}^{k+1}-t_{e,j^{+},c}^{k}\geq \Delta T_{c}$$

Next, the arrival time of the available gap at each conflict point is calculated. The available gap between vehicle k and vehicle k+1 in direction $d^{+}$ moves with the vehicles, and its arrival time at each conflict point is shown in Equation (16).(16)$$t_{e,d^{+},c+1}^{k}=t_{e,d^{+},c}^{k}+\frac{l_{d^{+},c,c+1}+w}{v}$$

Further, the duration of the available gap is calculated:
①For the first conflict point, if vehicle
k+1
has already reached the guidance speed at time
$t_{e,d^{+},1}^{k}$, then the duration of the available gap is as shown in Equation (17).(17)$$t_{b,d^{+},1}^{k+1}-t_{e,d^{+},1}^{k}=\frac{l_{b,d^{+},1}^{k+1}-l_{d^{+},1}^{k+1}(t_{e,d^{+},1}^{k})}{v}$$②For the first conflict point, if vehicle
k+1
has not reached the guidance speed at time
$t_{e,d^{+},1}^{k}$, then the duration of the available gap is as shown in Equation (18).(18)$$t_{b,d^{+},1}^{k+1}-t_{e,d^{+},1}^{k}=\left[\frac{v-v_{d^{+},1}^{k+1}(t_{e,d^{+},1}^{k})}{a_{d^{+},1}^{k+1}}\right]+\left[\frac{l_{b,d^{+},1}^{k+1}-l_{d^{+},1}^{k+1}(t_{e,d^{+},1}^{k})-\frac{v^{2}-{\left(v_{d^{+},1}^{k+1}(t_{e,d^{+},1}^{k})\right)}^{2}}{2a_{c}}}{v}\right]$$③For subsequent conflict points, Equation (19) holds true.(19)$$t_{b,d^{+},c+1}^{k+1}=t_{b,d^{+},c}^{k+1}+\frac{l_{d^{+},c,c+1}+w}{v}$$

If vehicle z from direction $h^{+}$ is to pass smoothly through conflict point c with direction $d^{+}$, then Equation (20) must be satisfied.(20)$$\left\{\begin{array}{l} t_{b,h^{+},c}^{z}+\frac{l+w}{v}\leq t_{b,d^{+},c}^{k+1}\\ t_{b,h^{+},c}^{z}\geq t_{e,d^{+},c}^{k}\\ t_{b,h^{+},c}^{z}=\left[\frac{l_{h^{+},c}}{v}\right]+\left[\frac{l_{b,h^{+},c}^{z}-\frac{v^{2}-{\left(v_{b,h^{+},c}^{z}\right)}^{2}}{2a_{h^{+},c}^{z}}}{v}\right] \end{array}\right.$$

On the basis of the above conditions, vehicle z can also pass smoothly through conflict point p, which is on the same path as conflict point c but conflicts with direction $d^{-}$, as shown in Figure 5. Therefore, Equation (21) must also be satisfied.(21)$$\left\{\begin{array}{l} t_{b,h^{+},c}^{z}+\frac{l+w}{v}+\frac{l_{h^{+},c,p}}{v}\geq t_{e,d^{-},p}^{u}\\ t_{b,h^{+},c}^{z}+\frac{2(l+w)}{v}+\frac{l_{h^{+},c,p}}{v}\leq t_{b,d^{-},p}^{u+1} \end{array}\right.$$

Due to the utilization of the available gap by vehicle z in direction $h^{+}$, this gap needs to be updated for the subsequent vehicle z+1, in the same direction $h^{+}$. Under this updated available gap, speed guidance is then realized for vehicle z+1. The corresponding update of the available gap for vehicle z+1 is shown in Equation (22).(22)$$\left\{\begin{array}{l} t_{e,d^{+},c}^{k}=t_{b,h^{+},c}^{z}+\frac{l+w}{v}+T_{safe}\\ t_{e,d^{-},p}^{c}=t_{b,h^{+},c}^{z}+\frac{2(l+w)}{v}+\frac{l_{h^{+},c,p}}{v}+T_{safe} \end{array}\right.$$

Based on Equation (22), it can be determined whether the updated gap still belongs to the available gap according to Equation (15).

Based on the above analysis, the overall algorithm for reducing search space is as follows:

Step 1: Add the traffic streams from all directions to set D, and from D, select streams that are non-conflicting with each other.

Step 2: Under the condition that safe car-following headways are met, assign initial guidance speeds to each vehicle in the selected traffic streams.

Step 3: Add the selected traffic streams to set G and remove them from set D.

Step 4: Use Equation (15) to determine if available gaps exist at the conflict points between the streams in set G and streams from other directions.

Step 5: Use Equation (16) to calculate the start time of the available gaps at each conflict point for the traffic streams in set G.

Step 6: Use Equations (17)–(19) to calculate the duration of the available gaps at each conflict point for the traffic streams in set G.

Step 7: From set D, select another set of mutually non-conflicting streams. Use Equations (20) and (21) to calculate the constraints imposed by the available gaps of the streams in G on the speeds of vehicles in this newly selected set of streams.

Step 8: Use Equation (22) to update the available gaps.

Step 9: Obtain the guidance speeds for each vehicle in the newly selected streams that satisfy the non-conflicting constraints. Remove the newly selected streams from set D and add them to set G.

Step 10: If set D is empty, terminate the algorithm; otherwise, proceed to the next step.

Step 11: From the set of all conflict points at the intersection, remove the conflict points between the traffic streams in set G. Return to Step 4.

## 5. Improved Double-Layer Multi-Population Particle Swarm

To solve the aforementioned model, this paper utilizes the particle swarm optimization (PSO) algorithm. PSO has the advantages of having few parameters, fast convergence speed, and being easy to implement. However, it also suffers from issues such as low convergence accuracy and a tendency to fall into local optima. Based on this, this paper designs an improved double-layer multi-population particle swarm algorithm to enhance the population diversity of particles during the search process and to improve convergence accuracy. The algorithm steps are as follows:

Step 1: The lower-layer sets have m populations, the number of particles in each population is the same, n particles are randomly generated within each population, and the initial positions of these n particles in their population are initialized as $(c_{l}^{k},t_{l}^{k}),k=1,2,\ldots ,K_{\Delta T}$. The dimension of each particle is the same as the dimension of the search space, and at the same time, the initialized global best position of each population is obtained. Here, $K_{\Delta T}$ represents the total number of vehicles entering the control zone within the time window ΔT; $c_{l}^{k}$ represents whether vehicle k changes lanes and the direction of the lane change. $c_{l}^{k}$ is represented in binary, where the 0th bit represents the lane change direction, with “1” representing a change to the right lane and “0” representing a change to the left lane; the 1st bit represents whether to change lanes, with “1” representing changing lanes and “0” representing not changing lanes. $t_{l}^{k}$ represents the time at which vehicle k enters the intersection.

As known from the particle’s vector representation, its position consists of two types of variables: continuous and binary. Among them, $t_{l}^{k}$ is a continuous variable, and a traditional update formula is used for its position update, as shown in Equation (23). $c_{l}^{k}$ is a binary variable, and the sigmoid function is used for its position update, as shown in Equation (24). When updating $c_{l}^{k}$, it is first necessary at the decision-making instant to determine whether vehicle k meets the conditions for a lane change and which directions it can change to. If vehicle k does not meet the conditions for a lane change, the corresponding $c_{l}^{k}$ is set to “00” and is no longer updated. If vehicle k meets the conditions but can only change to one adjacent lane (left or right), only the 1st bit of the corresponding $c_{l}^{k}$ can be updated. If vehicle k meets the conditions and can change to both adjacent lanes (left and right), then both bits of the corresponding $c_{l}^{k}$ can be updated.
(23)$$\left\{\begin{array}{l} v_{l}^{k}(t+1)=\omega *v_{l}^{k}(t)+c_{1}*rand*\left(pbest-t_{l}^{k}(t)\right)\\ +c_{2}*rand*\left(gbest-t_{l}^{k}(t)\right)\\ t_{l}^{k}(t+1)=t_{l}^{k}(t)+v_{i}^{k}(t+1) \end{array}\right.$$(24)$$\left\{\begin{array}{l} u_{i}^{k->a}(t+1)=\omega *u_{i}^{k->a}(t)+c_{1}*rand*\left(pbest-c_{l,i}^{k->a}(t)\right)\\ +c_{2}*rand*\left(gbest-c_{l,i}^{k->a}(t)\right)\\ c_{l,i}^{k->a}(t+1)=\left\{\begin{array}{l} 1,rand<Sigmoid\left(u_{i}^{k->a}(t+1)\right)\\ 0,otherwise \end{array}\right.\\ Sigmoid\left(u_{i}^{k->a}(t+1)\right)=\frac{1}{1+e^{-u_{i}^{k->a}(t+1)}} \end{array}\right.$$

In Equation (24), $c_{l,i}^{k->a},u_{i}^{k->a}$ represent the a−th bit of the binary position and velocity, respectively.

Step 2: The fitness of each particle in each population, $D_{sum}$, is calculated separately. Considering the constraint conditions in the model, and to accelerate the algorithm’s search for the optimal speed, this paper combines the constraint functions into a single penalty term [20]. This term is added to the original objective function to guide the iterations toward the feasible region. The penalty function is explained by the following formula:(25)$$D_{sum}(c_{l}^{k},t_{l}^{k},M)=D_{sum}(c_{l}^{k},t_{l}^{k})+Mp(c_{l}^{k},t_{l}^{k})$$
where $D_{sum}(c_{l}^{k},t_{l}^{k})$ is the objective function and $Mp(c_{l}^{k},t_{l}^{k})$ is the penalty term.

Step 3: Within each of the lower-layer populations, if the fitness value of the current particle is better than the fitness value of the global best solution, then the global best position is updated.

Step 4: Determine if the current running time step is a multiple of the set time step k. If it is, perform a one-to-one random crossover and exchange of particles between the lower-layer populations. The number of exchanged particles is set to $n_{1}$, as shown in Figure 6.

The random matching is calculated as follows:(26)$$\begin{array}{l} mate(i)=j,if\frac{j-1}{num_{no\_mate}-1}\leq a_{2}<\frac{j}{num_{no\_mate}-1},\\ j\in [1,2,\ldots ,num_{no\_mate}-1] \end{array}$$
where mate(i) is the population number matched with population i; j is the sequentially re-assigned number for the unmatched populations (not including population i); $num_{no\_mate}$ is the number of unmatched populations; and $a_{2}$ is a randomly generated number between 0 and 1.

Step 5: Upper-layer particle swarm optimization: The composition of the upper-layer particle swarm is updated in each crossover and exchange period. The best fitness value from each lower-layer population is compared with the worst fitness value of each particle in the upper layer. If a lower-layer best fitness value is better than an upper-layer worst fitness value, the particle corresponding to that best fitness value from the lower-layer population replaces the particle corresponding to that worst fitness value in the upper-layer population, as shown in Figure 7.

Step 6: Using the Orthogonal Experimental Design method combined with Gaussian perturbation, a virtual particle swarm is constructed for the best particle in the upper-layer population to perform local exploration for the optimal value (as shown in Figure 8). The other particles within the upper-layer population undergo iterative evolution using the Variable Neighborhood Search method. If the optimal value from the virtual particle swarm’s local search is still the population’s best when compared to the optimal value obtained from the iteration of the other particles, the local exploration continues, and simultaneously, the historical best particle from the virtual swarm replaces the worst particle in the main population. Otherwise, the particle corresponding to the optimal value from the other particles becomes the new population best, and at the same time, the best particle from the virtual swarm replaces the worst particle in the main population.

Step 7: The algorithm stops iterating after reaching the maximum number of iterations or satisfying the termination criteria, at which point it outputs the global best position found among all populations. If the termination conditions are not met, the algorithm returns to Step 2.

## 6. Model Solution, Validation, and Comparative Analysis

To further validate the optimization performance of the proposed model, simulation experiments were conducted. The initial conditions for the experimental environment were set as follows:(1)The experimental scene is a standard four-way intersection. The road sections in all directions are two-way, four-lane roads. Each approach has two inbound lanes: the inner lane is a dedicated left-turn lane, and the outer lane is a dedicated straight-through lane.(2)The traffic flow rate for each approach direction is 500 veh/h.(3)Vehicles from the North, East, South, and West approaches are marked in green, red, yellow, and blue, respectively. The total simulation duration is 1800 s.(4)While satisfying the constraint of Equation (6), the time window ΔT
is set to 3 s.

The specific simulation parameter settings for this experiment are shown in Table 2. Among them, due to the speed limit on the road section, we set the maximum speed of buses traveling on the road section to be the same as that of general vehicles. Meanwhile, the bus arrivals are set to be uniformly distributed.

After applying the speed guidance algorithm proposed in this paper, when each vehicle enters the control zone, the optimal guiding speed will be automatically calculated. The specific control method is to use the function “traci.vehicle.setSpeed(vehID, guide_speed)” from the TraCI API development interface of SUMO to control the speed, the real-time motion simulation of all vehicles is shown in Figure 9:

A satellite image of the selected four-way intersection is shown in Figure 10.

The space–time diagrams of the trajectories for the first 20 vehicles from each direction are shown in Figure 11:

To comprehensively evaluate the performance of the method proposed in this paper, three representative vehicle control methods were selected as comparative benchmarks:Traditional Signal Control: This adopts the actuated signal timing scheme from reference [28].First-In-First-Out (FIFO) Strategy: As the most fundamental method among signal-free control strategies, this approach determines the right-of-way based on the chronological order of vehicles entering the control zone. FIFO is simple to operate and does not require complex optimization calculations. It is widely used as a baseline for performance evaluation in autonomous driving research and can provide a lower-bound reference for control effectiveness.A Cooperative Driving Strategy for CAVs at Signal-free Intersections Based on Selection Sort: This is an optimization-driven control method proposed by reference [29] that achieves vehicle cooperation through a double-layer architecture. The dynamic sequencing layer constructs a weight matrix based on conflict resolution costs and uses an improved selection sort algorithm to adjust the passing sequence in real time, reducing the algorithm’s time complexity to the order of O (n log n). The trajectory planning layer generates safe trajectories by incorporating vehicle kinematic constraints and employs Rolling Horizon Optimization (RHO) technology to achieve real-time responses at the 50 ms level.

The three types of methods mentioned above represent the primary technical routes of “traditional signal control,” “rule-driven signal-free control,” and “dynamic-optimization-based signal-free control,” making them, respectively, highly representative and comparable. By comparing against these methods, the performance advantages of the algorithm proposed in this paper in terms of delay control, computational efficiency, and other aspects can be comprehensively assessed.


**1. Comparison of Average Delay Time**


To compare the average delay time under high, medium, and low traffic flow conditions, the flow rates were set to 200, 300, 400, 500, 700, and 800 veh/h, respectively. During the simulation comparison experiment, all parameters other than the traffic flow rate are kept constant, including vehicle parameters, car-following models, network configuration, and the proportion of buses and CAVs. Figure 12 shows the comparison of average delays for the four algorithms under different traffic volumes. The analysis indicates that the proposed method achieves the lowest average delay, significantly outperforming the traditional signal control method and others, with an improvement ranging from 56.3% to 81.3%.


**2. Comparison of Algorithm Computational Efficiency**


To compare the computational performance metrics of the algorithm proposed in this paper with others, comparative experiments were conducted on a workstation equipped with a Windows 11 64-bit operating system. The workstation’s processor is an Intel(R) Core(TM) i7-8700 CPU @ 3.20 GHz, with 32 GB of RAM and a 1 TB solid-state drive. Figure 13 presents a comparison of the computational efficiency of the different algorithms. The analysis reveals that while the FIFO algorithm consumes the least resources and has the fastest computation time, the result of its objective function is not globally optimal. The selection sort algorithm consumes the most resources and has a longer running time. The computational efficiency of the algorithm proposed in this paper is close to that of the FIFO algorithm.

The simulation experiment results demonstrate that the model proposed in this paper significantly reduces the average vehicle delay time by optimizing guidance speeds. It achieves a favorable balance between computational resource consumption and algorithmic efficiency, effectively proving the validity of the proposed speed guidance model.


**3. Speed Guidance Control Considering Relative Bus Priority**


A bus arrival rate was set to 30 vehicles per hour (veh/h). A comparative analysis was conducted on the per capita delay metric under two scenarios: speed guidance considering relative bus priority $\mu_{1}=1.5,\mu_{2}=30$ and speed guidance without considering bus priority $\mu_{1}=1.5,\mu_{2}=1.5$. The per capita delay within the simulation time was calculated based on an estimated passenger occupancy of 1.5 persons per general automated vehicle and 30 persons per bus. The simulation results are shown in Figure 14 (in the figure, the simulation time window indicates the period during which a bus enters the control area of the simulated road section). As can be seen from Figure 14, the average per capita delay for the scenario considering relative bus priority is 2.02 s, while the per capita delay for the scenario without considering bus priority is 2.49 s. The method considering relative bus priority reduces the per capita delay metric by 18.95% compared to the one that does not consider bus priority.

A further comparative analysis was conducted on the per capita delay metric between the relative bus priority and absolute bus priority speed guidance strategies. The per capita delay within the simulation time was calculated for each case, with the results shown in Figure 15. As can be seen from the figure, the relative bus priority strategy tends to be closer to the overall traffic optimum for the intersection compared to the absolute bus priority strategy. Under the set vehicle arrival conditions, the average per capita delay for relative bus priority is 2.02 s, while for absolute bus priority, it is 2.29 s. The former reduces the per capita delay metric by 11.79% compared to the latter.

## 7. Conclusions

This paper addresses the issue of vehicle control at signal-free intersections in the context of autonomous driving. First, grounded in an intersection coordinate system, a conflict area analysis model was constructed based on the coordinates of key conflict points and vehicle trajectories. Subsequently, considering the set of vehicles within an optimization time window that influences decision-making, an optimal scheduling model for automated vehicles at signal-free intersections with bus priority was designed, using vehicle guidance speeds on road sections and lane-changing vehicles as decision variables. At the same time, to avoid the practical limitations caused by excessively low guidance speeds and the problem of combinatorial explosion from an increased number of decision variables, the guidance of road section speeds and vehicle intersection entry times were unified. To obtain optimized solutions more efficiently, we designed an enhanced double-layer multi-population particle swarm optimization algorithm. Additionally, a search space reduction method based on available gaps for conflict point traversal constraints was proposed to improve the algorithm’s search speed in sparse spaces.

Simulation comparison results against control algorithms such as “traditional signal control,” “rule-driven signal-free control,” and “dynamic-optimization-based signal-free control” show that the method proposed in this paper can reduce vehicle delay by a maximum of 56.3–81.3%. Furthermore, a comparison between the control model considering relative bus priority and one without shows that the former reduces the per capita delay metric by 18.95% relative to the latter. These results effectively demonstrate the validity of the proposed method and can provide support for enhancing the efficiency of vehicle optimal scheduling at signal-free intersections in the context of developing autonomous driving technology.

Due to space limitations, this paper did not elaborate on the impact of passengers’ maximum tolerance time or the size of the time window on the optimization results; these issues can be addressed in subsequent research. Additionally, this paper assumes a fully automated driving environment, which to some extent limits the applicability of the proposed method. Future work can be extended to research on optimal vehicle-scheduling at signal-free intersections in mixed-traffic environments with both human-driven and automated vehicles. Finally, to strengthen the connection between the control model and practical applications, further research can be conducted on optimal control models for automated vehicles at signal-free intersections under the influence of pedestrians and non-motorized vehicles, as well as in non-fully automated environments.

## Figures and Tables

**Figure 1 sensors-25-05438-f001:**
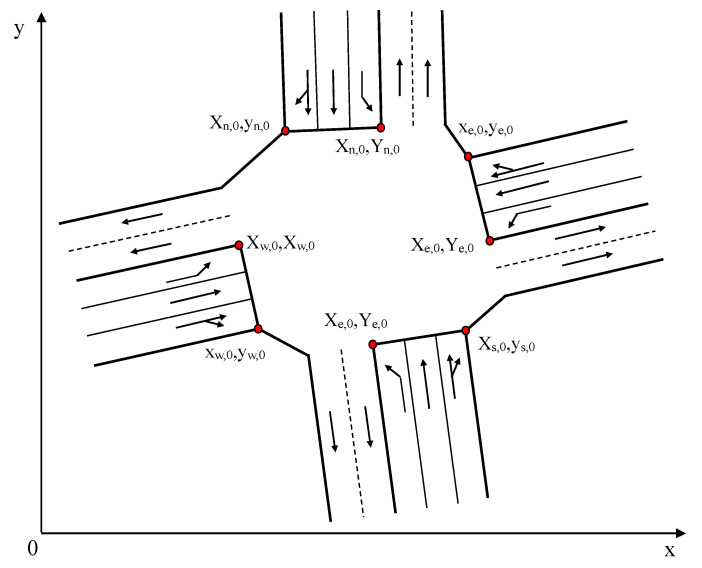
Diagram of problem description.

**Figure 2 sensors-25-05438-f002:**
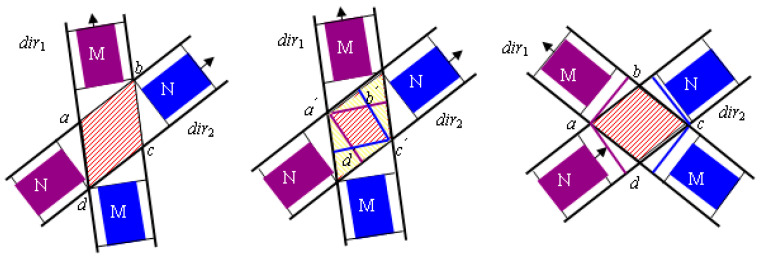
Diagram of conflict area analysis (straight and straight).

**Figure 3 sensors-25-05438-f003:**
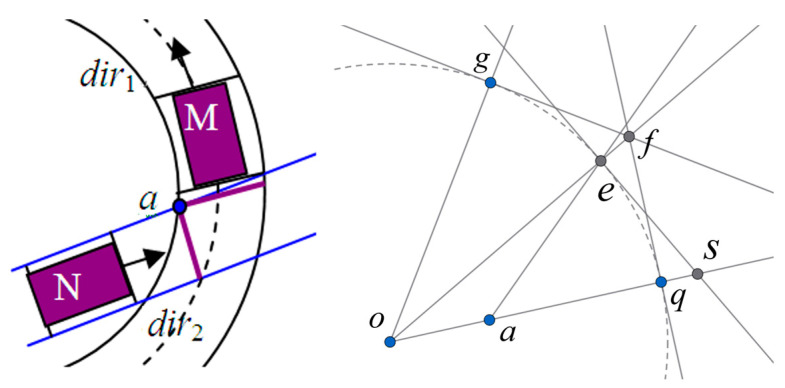
Diagram of conflict area analysis (straight and turning).

**Figure 4 sensors-25-05438-f004:**
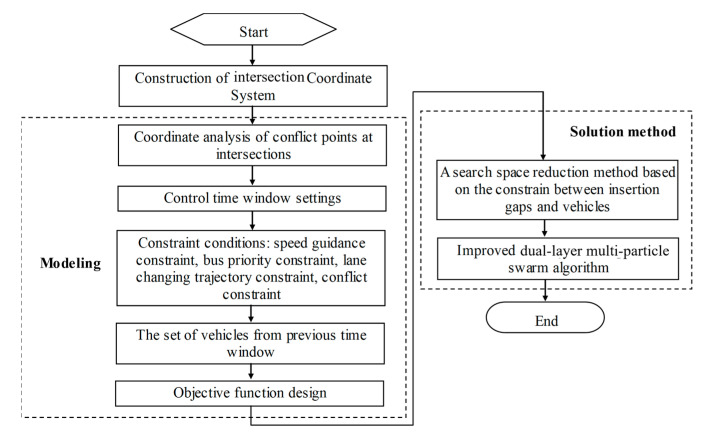
Overall flowchart of the control method.

**Figure 5 sensors-25-05438-f005:**
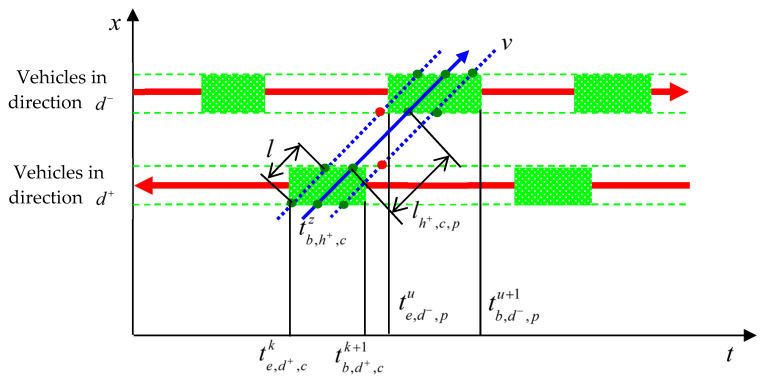
Constraint on conflict point traversal by the available gap.

**Figure 6 sensors-25-05438-f006:**
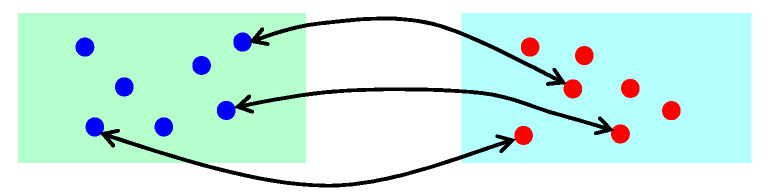
Schematic diagram of particle exchange between populations.

**Figure 7 sensors-25-05438-f007:**
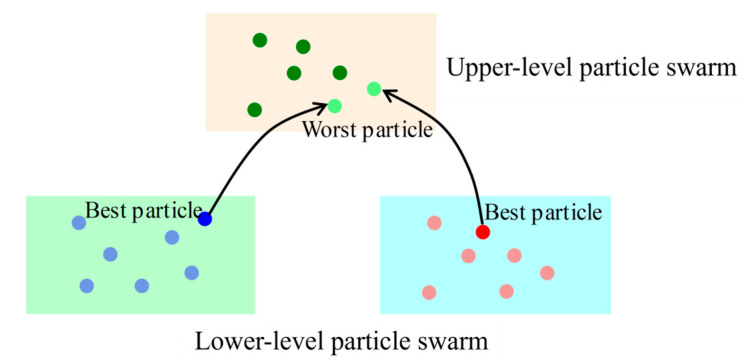
Particle exchange between upper level and lower level.

**Figure 8 sensors-25-05438-f008:**
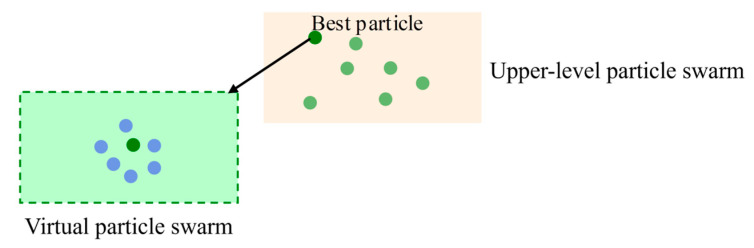
Virtual particle swarm.

**Figure 9 sensors-25-05438-f009:**
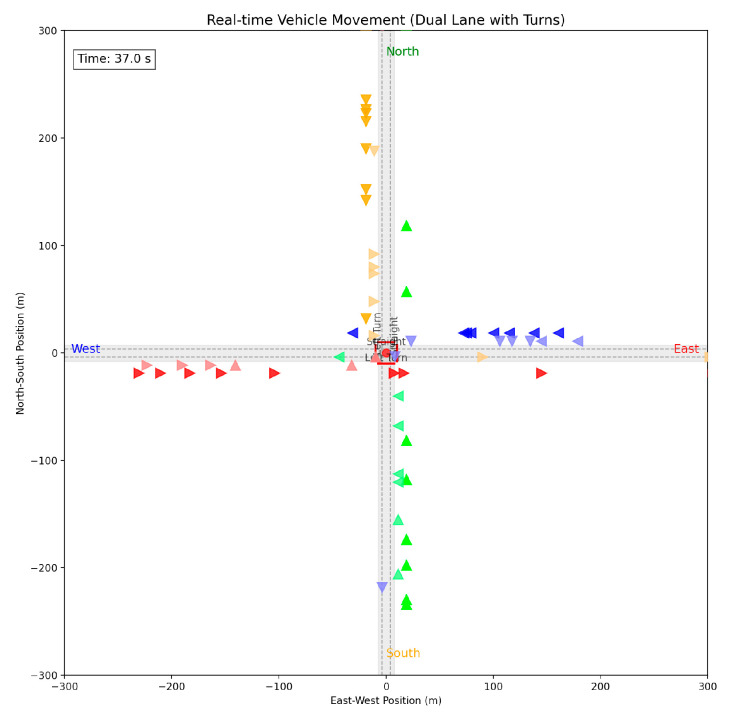
Real-time motion simulation of vehicles in the four-way intersection experimental scenario.

**Figure 10 sensors-25-05438-f010:**
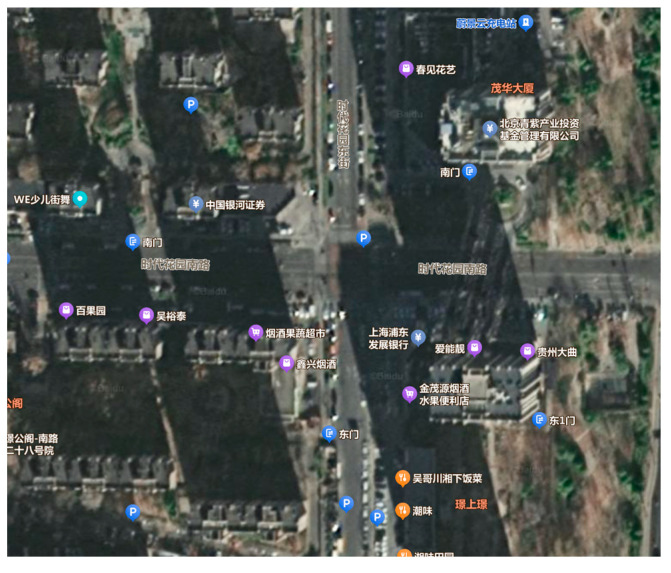
A satellite image of the four-way intersection.

**Figure 11 sensors-25-05438-f011:**
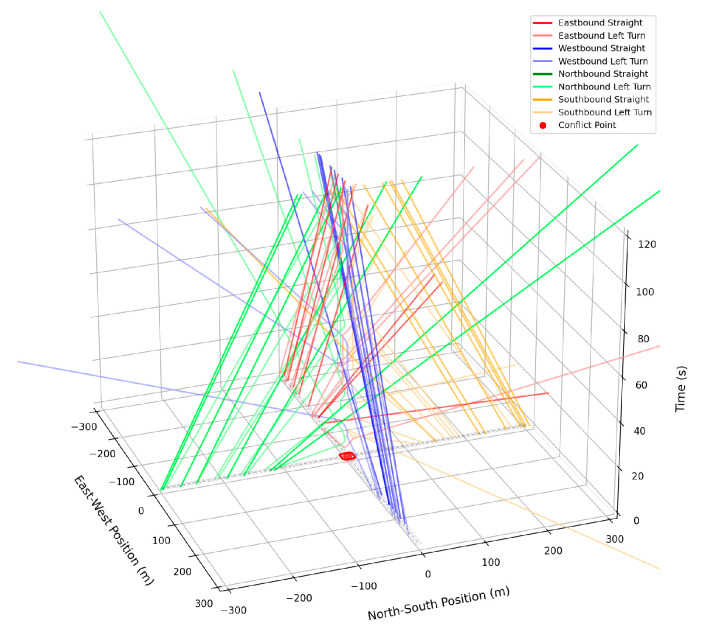
Spatiotemporal trajectories of vehicles under speed guidance.

**Figure 12 sensors-25-05438-f012:**
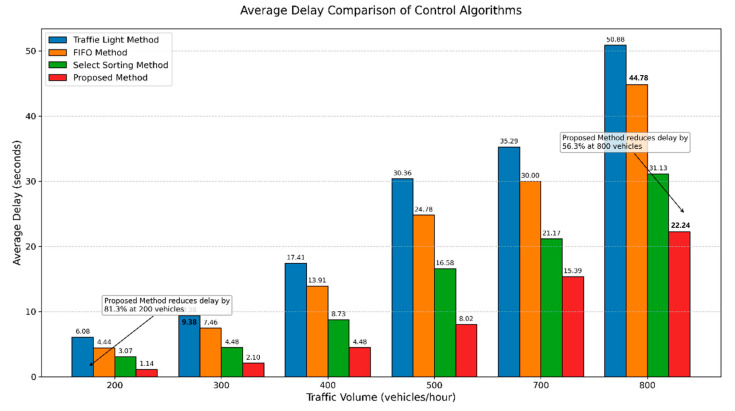
Comparison of delay times for four different control algorithms under various traffic flow conditions.

**Figure 13 sensors-25-05438-f013:**
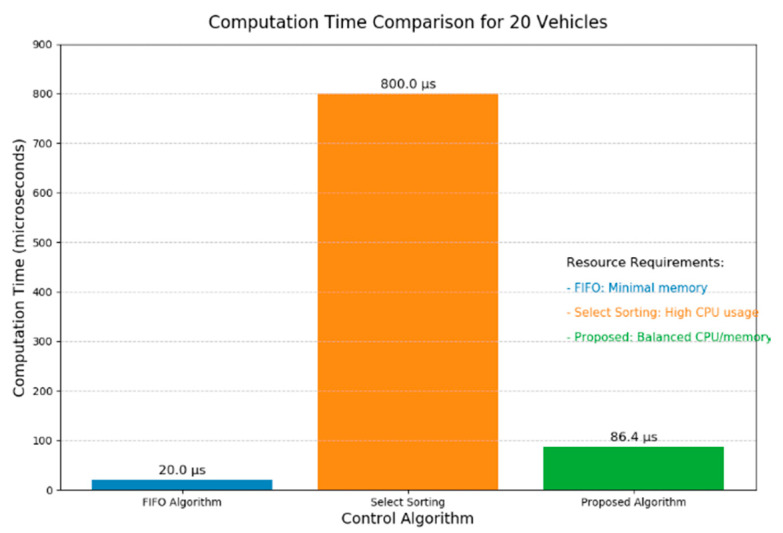
Comparison of computational performance metrics for different algorithms.

**Figure 14 sensors-25-05438-f014:**
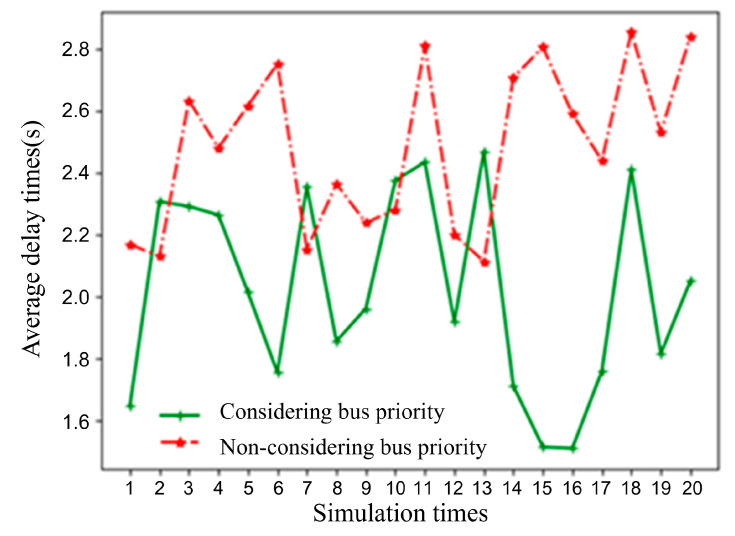
Comparison of per capita delay between the proposed model and the model without bus priority.

**Figure 15 sensors-25-05438-f015:**
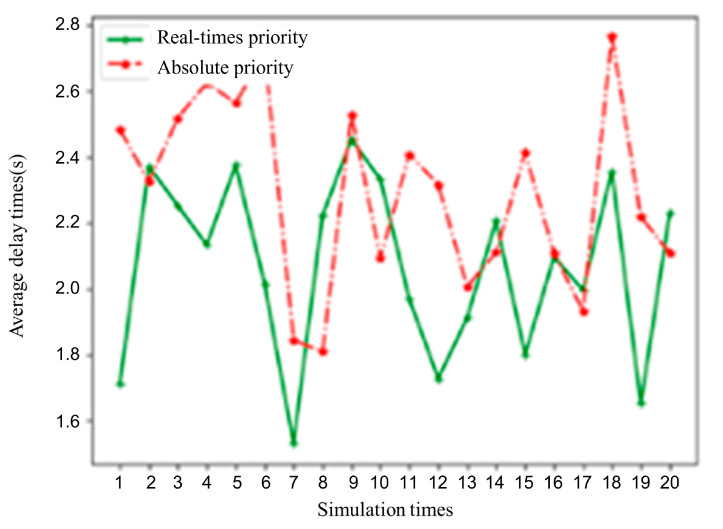
Comparison of per capita delay between the proposed model and the absolute bus priority speed guidance model.

**Table 2 sensors-25-05438-t002:** Experimental parameter settings.

Parameter	Value
car-following models	IDM
calibration of headway	1.6 s
warm-up durations	300 s
steady-state durations	1800 s
maximum acceleration	2.5 m/s^2^
maximum deceleration	1.8 m/s^2^
maximum speed limits	13.88 m/s
minimum guiding speeds	5.55 m/s
length of control zones	200 m
comfortable acceleration	1.2 m/s^2^
length of general vehicles	5 m
length of buses	12 m

## Data Availability

The original contributions presented in this study are included in the article.

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
