# Peer review of "An Optimal Vehicle-Scheduling Model for Signal-Free Intersections Considering Bus Priority in a Connected and Automated Vehicle Environment"

_sensors, 2025, doi:10.3390/s25175438_

Round 1
Reviewer 1 Report
Comments and Suggestions for Authors
The paper proposes an optimization model for unsignalized intersections that coordinates speed, lane changes, and entry timing in CAVs using an enhanced dual-layer multi-swarm PSO, while incorporating public transit priority. Although the paper is technically sound and its main formulations are well structured, the evaluation of the proposal is quite limited. Below, I outline my main concerns.
1) The study presents a brief and outdated review of related work. Furthermore, although the review highlights key limitations in autonomous intersection control, the authors fail to conclude how their approach addresses these gaps. The authors are encouraged to conduct a more in-depth state-of-the-art review, focusing on recent approaches, and to make an effort to contrast their solution both qualitatively and quantitatively against existing methods.
2) Section 3: “Control Model” presents the formulation of the control time window and its feasibility conditions, along with five key constraints, covering speed guidance, bus priority, lane-change trajectories, vehicle set continuity, and conflict avoidance, that together ensure consistent coordination of speeds and accelerations for intersection control. Nevertheless, the paper does not describe how these conditions are integrated into a car-following model suitable for replicating autonomous driving behavior, despite briefly stating that the proposal considers the CACC model. It is necessary to specify the CACC model rules that can be modified and to clearly illustrate how the proposed approach is integrated into the model.
3) Section 4 introduces the Enhanced Dual-Layer Multi-Swarm PSO algorithm, which is intended to serve as the core of the proposed optimization framework. However, the development of this section is notably limited. The algorithm's characteristics are only briefly outlined, and fails to explain how this optimization strategy aligns with the control paradigms introduced in the introduction, namely, centralized, distributed, or hybrid architectures. This omission weakens the overall coherence between the methodological foundations and the broader system-level considerations. Moreover, an analytical evaluation of the algorithm is essential to establish its computational complexity and assess its efficiency, which directly impacts the feasibility of future deployment. This is particularly important given the nature and volume of the input data, which could impose a significant computational burden as the system scales. Without such analysis, the practical applicability of the proposed method in real-time or large-scale scenarios remains uncertain.
4) The paper’s main limitation relies in the way the proposal is evaluated. Hence, Section 5 describes an experiment based on the well-known SUMO framework, hiding important configurations. While some irrelevant details, such as the specifications of the computer used to run the simulations, are thoroughly reported, several critical aspects are entirely overlooked. The paper omits key simulation configurations, including the total duration of the simulation in terms of time-steps or elapsed time, the calibration parameters for traffic flow composition and vehicle dynamics (e.g., acceleration, deceleration, space gap, sigma, etc.), raod network warming periods, and any explanation of how the default car-following model (e.g. Krauss) is replaced or modified to implement the proposed constraints. Another critical omission concerns the lack of explanation regarding how the proposed ΔT window is implemented and how many such windows are executed throughout each simulation. Regarding the experimental design, it is noteworthy that the performance evaluation relies exclusively on comparisons against a “No Speed Guidance” scenario and an “Inductive Control” strategy based on fixed priority rules. However, it completely overlooks comparisons with other optimization-based approaches, including those critiqued in the related work section. In particular, without a broader simulation-based analysis, it is rather evident that any optimization algorithm would naturally outperform both baseline methods, thereby limiting the significance of the reported improvements. Another limitation lies in the traffic composition, which includes only autonomous vehicles, and is further restricted to a total of just 16. This limited scale raises concerns about the representativeness of the simulation. Given the intersection geometry illustrated in Figure 1, it is unclear what kind of real-world road network would plausibly operate with only 16 vehicles, including transit buses, within a meaningful simulation window. Without a robust and well-structured experimental analysis, it is difficult to assess the actual contribution of the proposed approach.
Reviewer 2 Report
Comments and Suggestions for Authors
This paper has optimized the vehicle scheduling at unsignalized intersections in CAV environments. In the study, it proposed a comprehensive conflict zone analysis model integrating key collision point coordinates and trajectory dynamics. Moreover, a multi-objective optimization model is also improved. Some problems still need to be sloved to make the paper better.
- In Introduction, What are the relevant control measures for non-signal-controlled intersections and what is the latest research progress? A detailed explanation should be provided.
- The assumption 1 does not seem like a hypothesis but merely a limitation of the research scope by the author.
- In page 5, line 155. What is the constraint condition (5)?
- The paper lacks an overall introduction to the control method and explains its working process. It can be considered to add a flowchart for control and management and provide certain explanations.
- In page 6, line 175-181. Although the process can be omitted, the corresponding results in different situations should still be given.
- When introducing the control model, The physical meaning behind the designed formula needs to be explained to a certain extent. So as to facilitate readers' understanding.
- 3.3.3 lane change trajectory constraints is a bit too qualitative. Is there a more quantitative standard.
- In section 5, The satellite image of the intersection should be provided.
- In the simulation, only 16 vehicles were considered, which is a little too simple. Analysis and verification should be conducted around the situation of a long period of time and the presence of more vehicles.
- The limitations of this paper also need to be more illustrated in conclusion.
Reviewer 3 Report
Comments and Suggestions for Authors
The authors of this paper present the problem of vehicle control at intersections without traffic lights in the context of autonomous driving. They also proposed an optimization scheduling model that combines road segment speed guidance and lane change decisions, and also takes into account bus prioritization strategies.
In the first section entitled Introduction, the authors included a short introduction and a review of the literature. The literature review is also very modest. I propose to separate these two parts. In the Introduction section, the authors should present their research in a broad context, explain why it is important, and should also include the main aim of the work, conclusions, and an outline of the article. It is also advisable to include research questions. I also recommend expanding the list of cited references and expanding the literature review, which should be included in the article as a separate section.
The authors also didn't present a discussion of the results in the article. The authors should discuss the results in relation to previous studies. In this section, they should clearly indicate the significance of their research results in relation to existing literature on the same area of ​​research. Of course, this literature should be cited by the authors. It would also be advisable to explain what research results may be applicable.
The conclusions presented by the authors are too general. In the conclusions, the authors should include information such as: answers to the research questions and an indication of the contribution of the presented results to the development of a given area of ​​research.
Technical notes:
The graphs in Figures 7 and 8 are illegal.
Reviewer 4 Report
Comments and Suggestions for Authors
Dear Authors,
I read your article on the optimization of unsignalized intersections with great interest. I have only a few minor comments and suggestions:
- I appreciate the well-written introduction in which you mentioned the approaches of several international authors. However, the list of references appears to be in an incorrect format – please consult with the editor.
- Please check Figure 1: each arm of the intersection includes a left-turn lane marked with a "left + straight" arrow, but in my opinion, it should be "left only." At least according to our intersection design standards, a vehicle in this lane should not drive straight.
- I would also suggest enlarging Figure 1, as the labels are not visible clearly.
- Similarly, Figure 2 appears to be in low resolution – this also applies to Figures 6, 7, 8, 10, and 11 (at least in the PDF version I reviewed) – please check it.
- Please thoroughly check all formulas and notations in your own interest. I am not capable of verifying every relationship in detail.
- Is Figure 4 from PTV Vissim? Did you use this software? What software is shown in Figure 5? If it is SUMO, it should ideally be introduced earlier in the text, together with the version used.
- Figure 9 – please use black font and black auxiliary lines.
- You have achieved excellent results clearly visible in the accompanying graphs.
- Lastly, I would like to share a philosophical remark – it will be exciting to one day sit in an autonomous vehicle and see it pass through a gap between cars on the main road at full speed. We will all know it's been carefully calculated, but it may still be terrifying for the passengers.
Thank you.
Round 2
Reviewer 1 Report
Comments and Suggestions for Authors
After reviewing the revised version of the manuscript, it is noted that the authors have made a significant effort to improve the quality of the paper. Some issues raised during the initial round of review have been addressed, particularly, the updated literature review and the more detailed specification of the dual-layer, multi-swarm particle swarm optimization (PSO) algorithm. Additionally, the authors have clarified certain misunderstandings, such as confirming that the proposed model does not incorporate the CACC car-following model. Nevertheless, the manuscript still presents several flaws, which are detailed below.
1) The authors are encouraged to adopt a simplified and more consistent notation to improve readability and reduce ambiguity. Overall, the model specification lacks cohesion, which hinders the fluency of the text. This issue is compounded by the use of symbols that are insufficiently explained and equations without enumeration. For example, line 183 introduces directions using the set {N, S, W, E}, where the capital letter N denotes “north”, however, N is later reused to refer to a specific vehicle. Similarly, in Figure 2, certain coordinates are labeled a, b, c, and d (in lowercase), which are later used to define segments such as bc or dc. Yet, the notation becomes inconsistent when a mix of lowercase and capital letters is subsequently used to denote lines or segments. Including a table or a summary of the notation would enhance the clarity of the mathematical presentation. This would help reduce ambiguity, particularly in the case of symbols involving superscripts and subscripts, which are difficult to interpret without a clear reference.
2) The details regarding the simulation calibration and configuration are still missing. Despite the additions of some brief details about the time step ΔT, the total simulation duration (1800 s), and the traffic flow rates, the simulation framework remains under-specified. Key parameters required for replicability in a microscopic traffic simulator such as SUMO are still missing. The manuscript does not disclose the car-following model used, or the duration of warm-up and steady-state periods for valid data extraction. The manuscript provides no information regarding model calibration against real-world data. The characteristics of the vehicles, such as acceleration, deceleration, length, maximum speed, driver imperfection (sigma), or minimum space gap, are not specified. Moreover, no distinction is made between different vehicle types. Although a bus arrival rate of 30 veh/h is mentioned, the simulation lacks the parameters to represent bus behavior and dynamics. Authors are encouraged to provide a detailed parametrization, particularly since CAVs are modeled through a framework that does not provide such a vtype by default.
3) One of the questions raised in the first round of review concerns how the proposed control model is integrated into the simulation engine. Since the authors clarify that the CACC car-following model is not used, it is assumed that the control logic may have been implemented programmatically using one of the SUMO’s APIs, for instance TraCI or sumolib. If this is the case, the authors are encouraged to provide detailed information on the implementation strategies adopted.
4) Section 6 presents a comparison against three vehicle control methods under varying traffic flow conditions (200 to 800 veh/h). However, the manuscript does not specify whether all methods share the same vehicle parameters, car-following models, or network configuration. It is also unclear whether the alternative strategies were implemented within the same SUMO simulation or if previously reported results were used. Furthermore, the comparison lacks information on whether the proportion of buses and CAVs was kept consistent across all evaluated strategies.
5) A minor issue regards the way the revised version was presented. Most of the text has been highlighted, making it difficult to identify which specific fragments were actually modified. It is recommended that the authors use strikethrough formatting to indicate deleted content and a distinct color to highlight newly added text, thereby improving the readability of the revision. Also, attaching a clean version of the manuscript without markup would be appreciated.
Reviewer 2 Report
Comments and Suggestions for Authors
All the problems have been improved.
Author Response
Thank you very much for your comments
Reviewer 3 Report
Comments and Suggestions for Authors
The authors took my suggestions into account. Although they didn't divide the text into two sections: introduction and literature review, they significantly expanded this section and expanded the number of references.
I have a few comments regarding the article. In Chapter 4, lines 410-431 contain the subsequent steps of the algorithm. They are placed immediately after Equation 22 and are preceded by no explanation of the algorithm.
Line 433 – the word "algorithm" is unnecessary.
In Chapter 6, the authors included a comparison of the proposed algorithm with other solutions. Unfortunately, they significantly reduced the case study presented in the first version of the article, using an example to test the performance of the proposed algorithm.
I suggest that the authors thoroughly review the article and refine the section presenting the algorithms used, as well as the research section presenting the results for the case study proposed by the authors and comparing it with other algorithms.
Round 3
Reviewer 1 Report
Comments and Suggestions for Authors
After reviewing the latest version of the paper, I have no further comments.
Reviewer 3 Report
Comments and Suggestions for Authors
The authors took my comments into account and answered my questions.